# Comprehensive Analysis of Metabolites and Biological Endpoints Providing New Insights into the Tolerance of Wheat Under Sulfamethoxazole Stress

**DOI:** 10.3390/ijms26094257

**Published:** 2025-04-30

**Authors:** Yong Yang, Jiangtao Jia, Tao Han, Heng Zhang, Yvjie Wang, Luying Shao, Xinyi Wang

**Affiliations:** 1Henan Joint International Laboratory for Crop Multi-Omics Research, School of Life Sciences, Henan University, No. 85 Jinming Road, Kaifeng 475004, China; 104752160097@vip.henu.edu.cn (Y.Y.); jjt0821@126.com (J.J.); xhzhang17@163.com (H.Z.); yujie.wang@henu.edu.cn (Y.W.); shaoluying@henu.edu.cn (L.S.); wangxinyi000831@163.com (X.W.); 2National Key Laboratory of Cotton Bio-Breeding and Integrated Utilization, Henan University, No. 85 Jinming Road, Kaifeng 475004, China; 3School of Horticulture and Landscape Architecture, Henan Institute of Science and Technology, Xinxiang 453003, China

**Keywords:** photosynthetic pigment, reactive oxygen species, antioxidant enzyme activity, metabolites, wheat, sulfamethoxazole

## Abstract

Metabolomics is a commonly used method to study the responses of organisms to environmental changes. However, the relationships between metabolites and biological endpoints still need further discussion. In this study, we exposed wheat seeds to sulfamethoxazole (0, 1, 10, 100 mg/L) for 5 days. The results show that sulfamethoxazole (SMX) had an inhibitory effect on wheat growth. It reduced shoot length, root length, shoot fresh weight, root fresh weight, chlorophyll content, and carotenoid content. At the same time, it increased the concentration of reactive oxygen species, the activity of superoxide dismutase, the activity of peroxidase, and the activity of catalase in the root. An orthogonal partial least squares analysis and correlation analysis were performed. SMX transformed five key metabolic pathways. Notably, certain metabolic alterations exhibited negative correlations with reactive oxygen species (ROS) accumulation and antioxidant enzyme activities (including superoxide dismutase (SOD), peroxidase (POD), and catalase (CAT)), while showing positive associations with root growth parameters (fresh weight and length). Conversely, other metabolic changes appeared to promote ROS generation and enhance antioxidant enzyme activities, consequently inhibiting root growth. These findings offer novel perspectives on the metabolic regulation of wheat’s stress response to SMX exposure.

## 1. Introduction

Entering the 21st century, the global demand and use of antibiotics have increased significantly. The application of antibiotics can be roughly divided into two parts: half is used to treat human bacterial diseases [1,2], and the other half is widely used in agriculture, food, and other fields, mainly concentrated in animals in the breeding industry [3]. China is the world’s largest agricultural antibiotic user [4]. The abuse of antibiotics in agriculture not only leads to an increase in the number of drug-resistant strains [5], but also spreads to humans through the food chain, thus destroying the human microecological balance and posing a potential threat to human health [6]. In addition, in agriculture and animal husbandry, antibiotics are widely used to control pests and diseases to protect crops and livestock [7]. Some antibiotics go directly into the environment, while others seep into the environment during fertilization through animal manure [8,9]. Antibiotics in the environment not only affect plant growth, but may also cause harm to the stability and development of the animal husbandry and aquaculture industry [10].

Metabolomics is widely used in the field of life sciences. It analyzes the types, structures, quantities, changes, and functions of metabolites in organisms, and comprehensively, quantitatively, and systematically evaluates the status and health status of living organisms [11,12]. In environmental biology, metabolomics studies the changes in organism metabolites in a specific environment to evaluate the impact of environmental factors on them and reveal the interaction relationships between living organisms and the environment [13,14]. This is a great significance for understanding ecosystem stability, predicting environmental change trends, and formulating effective environmental protection measures.

SMX can cause changes in microbial communities, increase the abundance of antibiotic resistance genes in the environment, and decrease the abundance of proteobacteria and key nitrogen-transforming genes [15,16]. Studies have shown that SMX has different degrees of impact on plant growth and development, including no obvious impact on promotion growth and inhibition [17,18]. In addition, SMX can also cause changes in the components such as photosynthetic pigments, reactive oxygen species, carbohydrates, and amino acids, resulting in changes in related metabolic pathways [17,19]. However, the changes in these substances are not uniform, and the results vary greatly, respectively [20,21,22]. This investigation was designed to achieve three primary objectives: (1) systematically assess the phytotoxic effects of SMX on wheat growth parameters; (2) decipher the toxicological mechanisms of SMX through an integrated analysis of metabolic profiles and physiological endpoints; and (3) identify SMX-induced metabolic reprogramming and establish critical connections between altered metabolic pathways and corresponding phenotypic responses.

## 2. Results and Discussion

### 2.1. Morphological Characteristics

An investigation was conducted to characterize the phytotoxicological impacts of SMX on Triticum aestivum (Figure 1). The results indicate that SMX had an inhibitory effect on wheat growth, and this effect became more pronounced with the increase in its concentration (Figure 1a,b and Appendix A). Under the treatments of 10 mg/L and 100 mg/L sulfamethoxazole, the shoot length of wheat was significantly lower than that of the control group. Additionally, after being treated with 1, 10, and 100 mg/L sulfamethoxazole, the shoot fresh weight, root fresh weight, and root length of wheat were all significantly lower than those of the control group. Specifically, compared with the control, the shoot length of wheat treated with 1, 10, and 100 mg/L sulfamethoxazole decreased by 12.99%, 51.24%, and 79.56%, respectively; the root length decreased by 51.68%, 66.84%, and 81.99%, respectively; the shoot fresh weight decreased by 7.02%, 43.69%, and 69.86%, respectively; and the root fresh weight decreased by 23.06%, 67.06%, and 84.06%, respectively. The study showed that SMX had a strong inhibitory effect on the growth of ryegrass [23], which is consistent with our research.

SMX led to a significant reduction in chlorophyll content in shoots, with a decrease of 10.56–91.20% (Figure 1c), and showed a more significant decrease with the increase in concentration. This result is consistent with earlier studies indicating that SMX reduces chlorophyll content in wheat and ryegrass [23,24]. In addition, compared to the control group, the carotenoid content of seedlings treated with different concentrations of SMX also showed a decreasing trend of 14.86–87.97%, and also showed a characteristic of increasing with the increase in concentration.

### 2.2. Reactive Oxygen Species (ROS)

It was found that the production of reactive oxygen species (ROS) in plant roots significantly increased after the application of sulfamethoxazole (Figure 2a). Specifically, ROS levels in 1, 10, and 100 mg/L sulfamethoxazole treatments were 15.81 times, 137.30 times, and 253.36 times of those of the control group, respectively. Extensive research has demonstrated that chloroplasts serve as a major site of reactive oxygen species (ROS) generation in plants, primarily due to the elevated oxygen levels associated with photosynthetic activity [13,25]. The experimental results further confirm that, compared to the control plants, the chlorophyll content of plants treated with this drug was significantly lower. In addition, the agent stimulated reactive oxygen species synthesis in plant roots, suggesting that reactive oxygen species may be generated through non-chloroplast pathways. It has been reported that ROS can increase through mitochondria–ROS channels under environmental stress [13,26]. Additionally, certain antibiotics have been shown to upregulate NOX4 gene expression, further enhancing ROS accumulation [27].

### 2.3. Antioxidant Enzyme Responses

Superoxide dismutase (SOD) is a critical enzyme in maintaining cellular redox homeostasis. In the present study, exposure to sulfamethoxazole at concentrations of 1 mg/L and 10 mg/L induced only marginal changes in SOD activity, suggesting a limited oxidative stress response at these levels (Figure 2b), whereas the application of 100 mg/L sulfamethoxazole solution significantly increased SOD activity. Among them, the increased rate of SOD activity in different treatment conditions was 22.85%, 35.50%, and 56.39%. The previous relevant literature also pointed out that this drug can effectively promote SOD activity [24,28]. As the core enzyme of plant antioxidant defense system, superoxide dismutase (SOD) plays a crucial role in the process of removing reactive oxygen species. Song et al. [25] showed that SOD could catalyze the disproportionation of superoxide anion (O^2−^) and convert it into hydrogen peroxide (H_2_O_2_) and oxygen (O^2^), thus effectively reducing the damage of plant cells caused by oxidative stress. Therefore, it can be speculated that, when subjected to this drug, wheat may activate a protective mechanism to mitigate the damage caused by free radicals.

As the concentration of sulfamethoxazole gradually increased, POD activity was also enhanced (Figure 2c). Although 1 mg/L of sulfamethoxazole did not cause a significant change in the activity of POD, a significant increase in POD activity was observed when 10 and 100 mg/L were administered. Among them, the proportion of POD activity in each treatment condition was 1.28 times, 2.46 times, and 2.69 times, respectively. Studies have shown that peroxidase (POD) plays a key role in plant defense responses by regulating lignin biosynthesis, ethylene production, and degradation of indole-3-acetic acid (IAA) in response to pathogen infection and mechanical damage [25]. Excessive production of reactive oxygen species (ROS) in plants will lead to the activation of the corresponding stress resistance system, and the elimination process of ROS requires the synergistic enhancement of SOD and POD [29,30].

At the same time, we also noticed that CAT activity increased significantly with the increase in SMX concentration (Figure 2d). Sulfamethoxazole with different concentrations can significantly enhance CAT activity, which is consistent with previous studies on the effect of this drug on CAT activity [28]. The CAT activity of 1, 10, and 100 mg/L sulfamethoxazole treatment was 1.30, 1.66, and 2.31 times that of control treatment, respectively. In the plant antioxidant defense system, catalase (CAT), as one of the key enzymes, synergizes with SOD and POD, and effectively reduces the toxic effect of reactive oxygen species (ROS) by catalyzing the decomposition of H_2_O_2_ into H_2_O and O^2^ [25]. However, regarding the effects of environmental stress on CAT activity, there are differences in research results. Cui et al. [31] reported that stress inhibited CAT activity, while Kachout et al. [32] observed the opposite trend. This divergence may result from the experimental environmental conditions or the specific differences of plant species, indicating that the stress response mechanism of CAT has complex regulatory characteristics.

### 2.4. Metabolite Profiling and Response Analysis

A metabolomic analysis revealed that root fresh weight, root length, ROS content, SOD, POD, and CAT activities served as a sensitive phenotypic indicator of SMX exposure. In addition, metabolomics technology was used to investigate the changes in the metabolites in plants. Specifically, we used gas chromatography and mass spectrometry to thoroughly analyze each sample and identified 58 metabolites [13]. Significant alterations were observed across multiple metabolite classes, particularly amino acids and organic acids (Figure 3). Hierarchical clustering classified the metabolic profiles into two distinct clusters, (1) control (C) and 1 mg/L SMX treatments and (2) 10 mg/L and 100 mg/L SMX treatments, suggesting a dose-dependent metabolic shift. The results show that high concentration of sulfamethoxazole could significantly change the internal metabolic activity of wheat cells. To systematically characterize stress-responsive metabolites, we employed a Venn diagram approach to identify both conserved and treatment-specific metabolic signatures across all experimental groups (C, 1, 10, and 100 mg/L SMX) (Appendix A). The four groups contained a total of 37 common components, of which the C and 100 mg/L groups each had 4 unique components.

Finally, to evaluate the potential relationships between various biochemical indicators in plant roots and biological endpoints, the orthogonal partial least square method was used for data analysis (Figure 4). Furthermore, variable importance in projection (VIP) analysis was conducted to quantify metabolite contributions to phenotypic endpoints. Notably, 33 metabolites with VIP scores >1 demonstrated significant associations with root fresh weight variation (Figure 4a), highlighting their potential as key regulators of SMX-induced growth inhibition. We further classified these metabolites according to their effect on root fresh weight. Among them, 26 metabolites had positive effects on root fresh weight (coefficients > 0), and 32 metabolites had negative effects (coefficients < 0). Thirteen specific metabolites, such as lactose, myo-inositol, xylose, and allose, had significant positive impact on root fresh weight (VIP values > 1, coefficients > 0). A total of 20 metabolites, such as sucrose, glucose, serine, and arabinopyranose, showed significant negative effects (VIP values > 1, coefficients < 0).

Next, we further analyzed the relationships between changes in root metabolite and other biological endpoints (Figure 4b–f). We found that 24, 29, 33, 35, and 25 metabolites had significant effects on ROS level, root length, SOD activity, POD activity, and CAT activity, respectively. The remaining metabolites had no significant effects with the measured parameters, including ROS level, root length, SOD activity, POD activity, and CAT activity (Figure 4b–f). The higher correlation between metabolites and root fresh weight, ROS level, root length, SOD activity, POD activity, and CAT activity, the higher their corresponding VIP values. Furthermore, metabolites showing positive correlations exhibited positive coefficients, while those with negative correlations displayed negative coefficients. These demonstrated that both VIP values and coefficients can effectively reflect the association between metabolites and various indicators (Appendix A). Taking glucose as an example, it showed strong negative correlations with root fresh weight (correlation: −0.946, VIP: 1.44776, coefficient: −0.03638) and root length (correlation: −0.975, VIP: 1.43377, coefficient: −0.0358). In contrast, it exhibited strong positive correlations with ROS level (correlation: 0.926, VIP: 1.35872, coefficient: 0.033544), SOD activity (correlation: 0.993, VIP: 1.44667, coefficient: 0.036105), POD activity (correlation: 0.950, VIP: 1.42652, coefficient: 0.035386), and CAT activity (correlation: 0.957, VIP: 1.39881, coefficient: 0.034604). These results indicate that VIP values and coefficients can clearly distinguish between the positive and negative correlations of metabolites with different parameters, further validating their reliability in metabolomic analyses.

It is worth noting that metabolites affecting root fresh weight and root length had positive effects on reactive oxygen species (ROS) levels and the activities of superoxide dismutase (SOD), peroxidase (POD), and catalase (CAT). In addition, metabolites that had positive effects on root fresh weight and root length had negative effects on ROS levels and SOD, POD, and CAT activities. Based on these findings, we developed a novel metabolic association model to (1) decipher the mechanistic links between specific metabolic perturbations and phenotypic responses and (2) establish an analytical framework for assessing phytotoxic effects under environmental stress conditions.

For differential metabolite screening, we applied a conservative threshold of two-fold change (FC ≥ 2 or FC ≤ 0.5) relative to control samples, ensuring the identification of biologically relevant metabolic alterations. According to MetaboAnalyst analysis, 1, 10, and 100 mg/L sulfamethoxazole significantly affected 8, 13, and 15 metabolic pathways in wheat roots, respectively. SMX acts on the following pathways: arginine biosynthesis; galactose metabolism; glycine, serine, and threonine metabolism; oxaloacetic acid and dicarboxylic acid metabolism; and starch and sucrose metabolism (Figure 5). SMX significantly perturbed key metabolic pathways, ultimately affecting wheat’s physiological state. As illustrated in Figure 6, comprehensive metabolic profiling revealed that SMX-induced alterations in amino acid metabolism were strongly associated with both root growth inhibition and ROS accumulation [33]. Previous studies have pointed out that the down-regulation or up-regulation of amino acid synthesis is an important manifestation of plant response to environmental stress [13,34], and multiple amino acids are involved in different metabolic pathways [31]. These results align with the established literature documenting stress-induced differential regulation of amino acid biosynthesis pathways, where certain amino acids accumulate while others decline under environmental stressors [34,35]. Glutaminase converts glutamine to glutamate, and glutamate dehydrogenase converts it to alpha-ketoglutaric acid. The intermediates produced from these reactions are used for enzymatic reactions for mitochondrial respiration and the production of adenosine triphosphate (ATP) [36]. At the same time, glutamate is a non-enzymatic antioxidant that regulates intracellular antioxidant morphology and participates in the composition of glutathione (GSH) [37]. Studies have shown that carbohydrates not only provide energy for plant growth, but also improve its ability to withstand environmental stress [38]. An increase in carbohydrate content was also observed in this study. Therefore, based on the analysis of metabolites and biological endpoints, we further discuss the mechanism of stress toxicity of sulfamethoxazole.

SMX exhibits moderate persistence, with degradation rates influenced by factors such as pH, organic matter content, and microbial activity. Prolonged SMX exposure enriches ARGs in soil microbiomes, contributing to the spread of multidrug-resistant bacteria [39,40]. The transfer of SMX and ARGs into the food chain via contaminated crops or water raises public health concerns. Additionally, the erosion of soil ecosystem services may compromise agricultural sustainability. SMX contamination in agriculture poses multifaceted ecological risks, from soil microbiome disruption to ARG dissemination. Future research should prioritize field-based studies to quantify long-term impacts and evaluate mitigation efficacy.

## 3. Materials and Methods

Wheat (*Triticum aestivum* L.) seeds (Henan Institute of Science and Technology, Xinxiang, China) underwent surface sterilization in 2% H_2_O_2_ for 15 min followed by triple-rinsing with ultrapure water. Sterilized seeds were then transferred to 4.5 cm diameter Petri dishes and exposed to SMX solutions at concentrations of 0 (control), 1, 10, and 100 mg/L (*n* = 3 replicates per treatment). The concentration of sulfamethoxazole was determined based on the environmental concentration of sulfamethoxazole and the concentration in related toxicological studies [21,39]. Cultivation occurred in controlled growth chambers under the following conditions: 16/8 h light/dark photoperiod (6000 lux illumination during light phase), with corresponding temperatures of 25 °C (light) and 18 °C (dark), maintained at 70% relative humidity. The experimental duration was 5 days. After that, the shoots were immersed in 8.0 mL etho-acetone-water mixed solution (9:9:2) for extraction for seven days, and then the absorbance of the extracted solution at 470 nm, 645 nm, and 663 nm was measured using a spectrophotometer, and the chlorophyll and carotenoid contents were calculated based on the absorbance data [33].

### 3.1. ROS Levels

Reactive oxygen species levels were detected using 2′,7′ -dichlorodihydroluciferin diester (DCFH-DA), which reacts to produce dichloroluciferin. It should be emphasized that, because DCFH-DA is non-polar, hydrophobic, and non-fluorescent, it is easy to enter cells. The specific operation steps were as follows: After cleaning the wheat shoots and roots three times, soak the wheat shoots and roots in 20 °C darkness for 40 min, and then rinse with ultra-pure water three times. The shoot tips and root tips were then observed and photographed using a fluorescence microscope (Olympus X71), and the relative fluorescence intensity was measured using Image J 1.43 software [13,33].

### 3.2. Antioxidant Enzyme

Following the 5-day treatment period, root samples were immediately harvested for biochemical and metabolomic analyses. For enzyme activity assays, fresh root tissues were homogenized in ice-cold 62.5 mM phosphate buffer (pH 7.4) and centrifuged at 15,000× *g* (4 °C, 10 min). The resulting supernatants were used to determine SOD, POD, and CAT activities following the spectrophotometric methods described by Han et al. [33]. Enzyme activities were quantified by measuring absorbance at characteristic wavelengths (SOD: 560 nm; POD: 470 nm; CAT: 240 nm) and calculated using established protocols.

### 3.3. Metabolites

For metabolomic profiling, harvested root samples were flash-frozen in liquid nitrogen and then ground in 2 mL of solution containing chloroform, water, and methanol (the ratio of 2:2:5), so that the cells were broken, separated at 4 °C by 11,000× *g* centrifugation, the above steps were repeated once. The mixed supernatant was then centrifuged for 5000× *g* for three minutes, followed by nitrogen drying and freeze-drying. Finally, the dried metabolites are mixed with methoxamine hydrochloride (20 g/L) and oscillated. After incubation at 30 °C for 90 min, 80 μL n-methyl-n-(trimethylsilyl)-trifluoroacetamide was added and incubated at 37 °C for 30 min in the same way, and finally injected into HP-5ms gas chromatographic column (30 m) (Agilent, Santa Clara, CA, USA) for derivative analysis. Metabolite detection was performed using non-targeted gas chromatography–mass spectrometry (GC-MS) analysis. The GC-MS system was operated under the following optimized parameters: injection port temperature: 230 °C, transfer line temperature: 250 °C, temperature program: initial hold at 80 °C for 2 min, followed by a 15 °C/min ramp to 325 °C. Compound identification was achieved by comparing mass spectra with the NIST 14.0 reference library (match threshold > 80%) [13]. To enable the metabolite content to be compared, total peak area normalization is performed on the data.

### 3.4. Statistical Analysis

SPSS 16.0 software was used for data analysis. To test the homogeneity of the variance, we used one-way ANOVA to identify significant differences with Dunnett’s C analysis (SOD, POD, and CAT activity) and LSD analysis (other indicators except SOD, POD, and CAT activity and metabolites) (*p* < 0.05). All experimental data are presented as mean ± standard deviation In addition, we used SIMCA 14.1 software for cluster analysis (Figure 4), MeV 4.9.0 software to data visualization (Figure 3), Cytoscape 3.9.1 software for the relationships between metabolites and biological endpoints (Figure 4), and MetaboAnalyst 6.0 software for differential metabolic pathways (Figure 5).

## 4. Conclusions

Using metabolomic profiling, this study systematically elucidates the phytotoxic effects of sulfamethoxazole (SMX) on wheat, establishing critical linkages between metabolic perturbations and phenotypic responses. Our integrated approach not only identified SMX-sensitive metabolic pathways, but also delineates their functional associations with key growth parameters and oxidative stress indicators. These findings reflect changes in metabolites induced by sulfamethoxazole. SMX exerted its effects by targeting multiple metabolic pathways, including the following: arginine biosynthesis; galactose metabolism; glycine, serine, and threonine metabolism; oxaloacetic acid and dicarboxylic acid metabolism; and starch and sucrose metabolism. This broad-spectrum activity allows for the antibiotic to interfere with essential biochemical processes in susceptible organisms. Notably, some metabolites that were negatively correlated with reactive oxygen species (ROS) levels and superoxide dismutase (SOD), peroxidase (POD), and catalase (CAT) activities were positively correlated with root fresh weight and root length. Some other metabolites may had adverse effects on root fresh weight and root length by increasing ROS levels and SOD, POD, and CAT activities. Therefore, this study on wheat deepens our understanding of the mechanisms of antibiotic-related toxic metabolic responses. In addition, we establish a link between root metabolite changes and biological endpoints to further elucidate the mechanism of action of antibiotic toxicity.

## Figures and Tables

**Figure 1 ijms-26-04257-f001:**
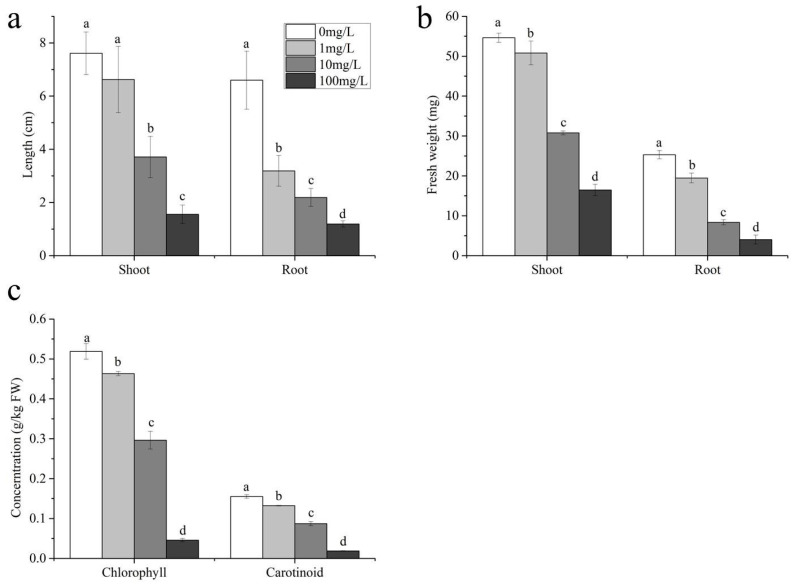
Morphological characteristics: (**a**) length; (**b**) fresh weight; (**c**) photosynthetic pigment. Different lowercase letters indicate significant differences found with the one-way ANOVA (LSD) (*p* < 0.05).

**Figure 2 ijms-26-04257-f002:**
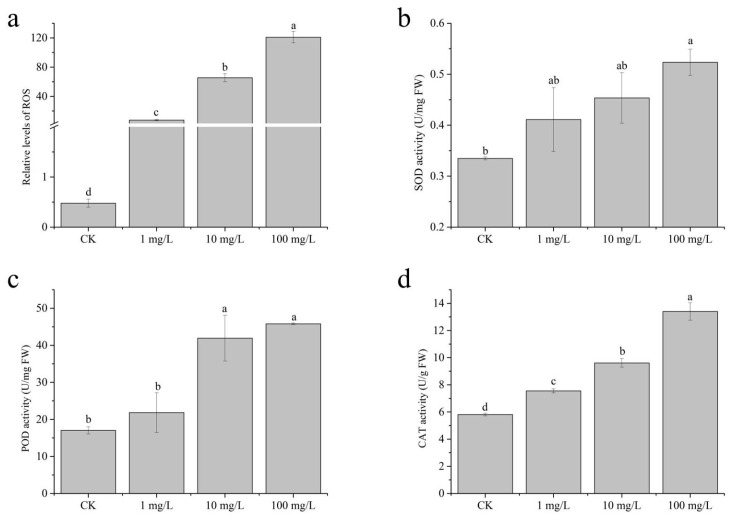
ROS and antioxidant enzyme responses: (**a**) ROS level; (**b**) SOD activity; (**c**) POD activity; (**d**) CAT activity. Statistical analysis with one-way ANOVA (ROS: LSD; other indicators: Dunnett’s) (*p* < 0.05) revealed significant variations among treatments, as indicated by different lowercase letters.

**Figure 3 ijms-26-04257-f003:**
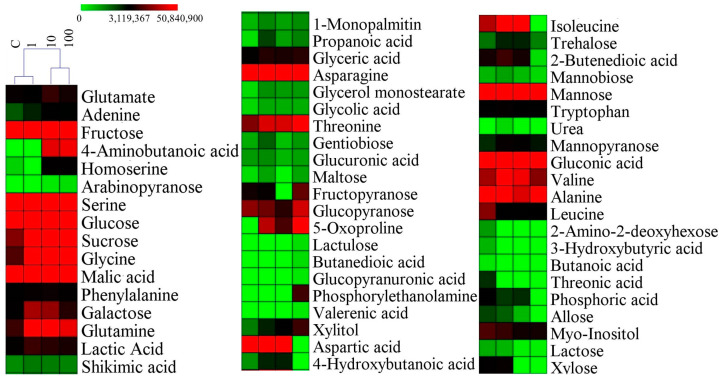
The relative contents of metabolites. MeV 4.9.0 software (Hierarchical Clustering) to data visualization. C = control (0 mg/L).

**Figure 4 ijms-26-04257-f004:**
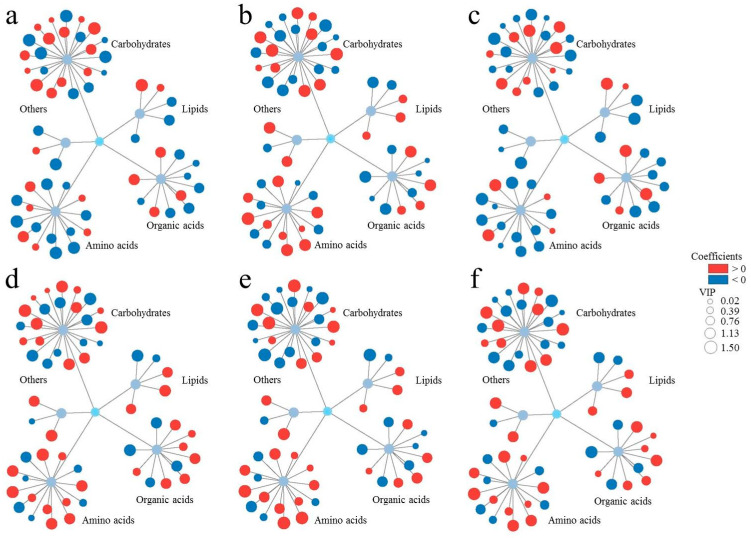
The comprehensive relationships between metabolic profiles and biological endpoints ((**a**): root fresh weight, (**b**): ROS level, (**c**): root length, (**d**): SOD activity, (**e**): POD activity, (**f**): CAT activity). SIMCA 14.1 software (Orthogonal Partial Least Square) for cluster analysis. Cytoscape 3.9.1 software to data visualization.

**Figure 5 ijms-26-04257-f005:**
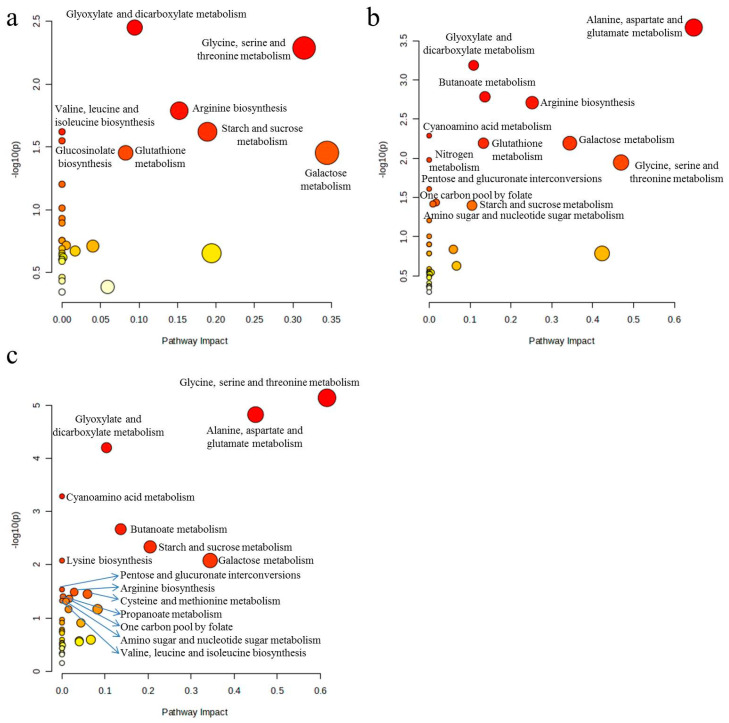
Differential metabolic pathways between control and sulfamethoxazole treatments: (**a**) C vs. 1 mg/L; (**b**) C vs. 10 mg/L; (**c**) C vs. 100 mg/L. C = control (0 mg/L). MetaboAnalyst 6.0 software (Fisher’s Exact Test) for differential metabolic pathways.

**Figure 6 ijms-26-04257-f006:**
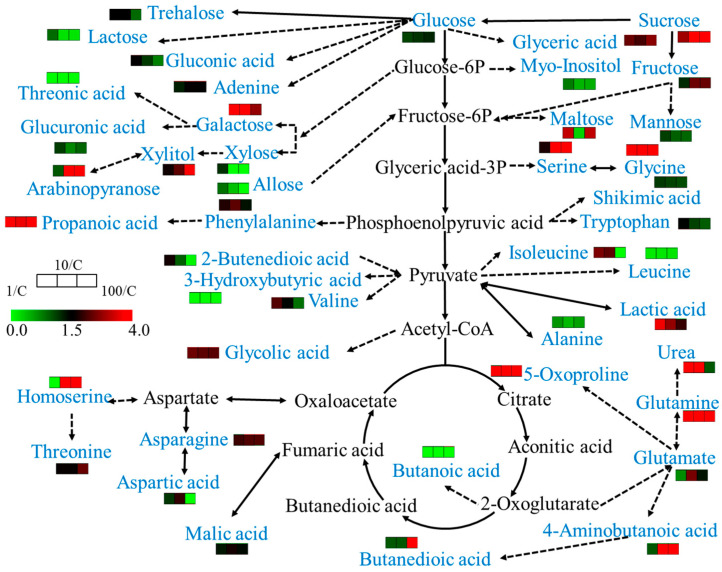
Metabolic pathways as affected by sulfamethoxazole treatment. Blue indicates metabolites revealing differential accumulation between sulfamethoxazole-treated and untreated plants.

## Data Availability

The main results and Appendix A have already been presented in the manuscript, and the original data can be obtained from the corresponding author on reasonable request.

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
