# Peer review of "Comprehensive Analysis of Metabolites and Biological Endpoints Providing New Insights into the Tolerance of Wheat Under Sulfamethoxazole Stress"

_ijms, 2025, doi:10.3390/ijms26094257_

Round 1
Reviewer 1 Report
Comments and Suggestions for Authors
dear authors
The manuscript requires significant revision before it can be considered for publication. Several sections lack clarity, critical analysis, and proper justification of methods and results. Additionally, the similarity index is 40%, which is unacceptably high and raises concerns about originality. The authors must ensure proper paraphrasing and citation throughout the text. Please refer to the attached document, where specific comments and suggestions have been provided. These must be carefully addressed to improve the scientific quality and integrity of the manuscript.
best regards

Author Response
Comments 1: The manuscript requires significant revision before it can be considered for publication. Several sections lack clarity, critical analysis, and proper justification of methods and results. Additionally, the similarity index is 40%, which is unacceptably high and raises concerns about originality. The authors must ensure proper paraphrasing and citation throughout the text. Please refer to the attached document, where specific comments and suggestions have been provided. These must be carefully addressed to improve the scientific quality and integrity of the manuscript.
Response 1: [Thank you very much for your question. We have perfected these contents. Please refer to manuscript.]
Comments 2: The article presents a 40% similarity index with previously published literature, which is unacceptably high for an original research manuscript. Such a level of textual overlap raises serious concerns about the manuscript’s originality and academic integrity. It is essential that the authors revise the text thoroughly to ensure proper paraphrasing, citation, and a clear distinction between their novel contributions and existing literature.
Response 2: [Thank you very much for your question. We have perfected these contents. Please refer to manuscript.]
Comments 3: should be reviewed and corrected: (1) The summary presents too many percentages and figures that make it difficult to read fluently; it would be better to summarize the general effects and leave the details for the body of the text. (2) A clearer statement of the objective of the study at the beginning and a stronger conclusion that highlights the relevance of the finding are missing. (3) The use of OPLS is mentioned, but there is no justification for choosing this technique or a clear explanation of its contribution.
Response 3: [Thank you very much for your question. We have perfected these contents. Please refer to manuscript.]
Comments 4: please improve the key words
Response 4: [Thank you very much for your question. We have made changes. Please refer to “key words”.]
Comments 5: The introduction lacks critical depth: it merely describes general facts without adequately contextualizing the environmental impact of sulfamethoxazole in real agricultural systems. There is no justification as to why the study is novel or what gaps in knowledge it addresses. The connection between metabolomics and specific effects on wheat is weak. Furthermore, the objective of the study appears to be a list and not a true scientific hypothesis.
Response 5: [Thank you very much for your question. We have perfected the method. Please refer to manuscript.]
Comments 6: It is suggested to reduce unnecessary repetition of results between text and figures, and to focus on interpreting the findings rather than simply describing them. There is a lack of critical discussion contrasting with previous studies or proposing possible physiological mechanisms. In addition, the use of the chosen concentration ranges is not justified. The section could benefit from clearer and more precise wording to highlight the novelty of the study.
Response 6: [Thank you very much for your question. We have perfected these contents. Please refer to manuscript.]
Comments 7: The section repeats data without elaborating on their biological interpretation; it merely describes percentage increases without discussing whether these changes are adaptive, toxic or part of a specific stress response. There is a lack of clear linkage of these enzymes to the observed growth damage. In addition, there is no discussion of whether these response levels are comparable with other studies, nor is the ecological relevance of the results evaluated.
Response 7: [Thank you very much for your question. We have perfected these contents. Please refer to manuscript.]
Comments 8: The section is excessively long, overloaded with figures and descriptions without a clear synthesis or critical discussion. The OPLS technique is abused without justifying its use compared to other statistical tools. The most relevant metabolic pathways are not highlighted and their physiological implications are not discussed. In addition, the biological interpretation is weak and repetitive, which dilutes the impact of the findings.
Response 8: [Thank you very much for your question. We have perfected these contents. Please refer to manuscript.]
Comments 9: The methodology lacks justification in the choice of sulfamethoxazole concentrations and number of replicates, which limits statistical robustness. No negative or positive controls for ROS are specified, nor are quality criteria for metabolomic data detailed. In addition, key information on validation of the analytical method and treatment of missing data is omitted.
Response 9: [Thank you very much for your question. We have perfected these contents. Please refer to “3. Materials and Methods”. We chose a solution with 4 concentrations and 3 replicates per concentration. To enable the metabolite content to be compared, total peakarea normalization is performed on the data.]
Comments 10: The conclusions are vague and reiterative; they do not clearly highlight the most relevant findings or their scientific or practical implications. There is a lack of a strong statement on the novelty of the study and its contribution to previous research. In addition, it avoids proposing future lines of research or potential applications, which weakens the closure of the work.
Response 10: [Thank you very much for your question. We have perfected these contents. Please refer to “4. Conclusions”.]

Reviewer 2 Report
Comments and Suggestions for Authors
Dear Authors,
Reviewer comments ijms-3554322
The manuscript entitled „Comprehensive analysis of metabolites and biological endpoints provides new insights into the tolerance of wheat under sulfamethoxazole stress“ represents a useful study aimed at an investigation of the effects of sulfamethoxazole treatment on wheat response at physiological and biochemical (metabolite) levels. I think that the manuscript provides interesting novel data on sulfamethoxazole effects on wheat physiological and biochemical characteristics which are worth publishing in International Journal of Molecular Sciences.
However, I have several comments on the present manuscript which are provided below.
In Materials and methods, the wheat plants used for the experiments have to be properly characterized by providing the information on the wheat cultivar/genotype used and the source of seeds by providing the institution (genebank, breeding company) name, city, and country.
In Materials and methods, appropriate description of methodological approaches for the presenetd data has to be added, i.e., the description of methodology used for the construction of endpoint graphs provided in Figure 4 has to be provided including the software or any other online tools used.
In Materials and methods, section on Statistical analysis, the kind of a post-hoc test of multiple comparisons following ANOVA analysis has to be specified. The same information, i.e., the kind of statistical test used for the determination of significant differences, has to be added to the legends to Figure 1 and Figure 2.
In Materials and methods, section 3.3. Metabolites, the basic information on gas chromatographic column including the company (manufacturer) has to be added.
In Results, I would recommend the authors to add some kind of multidimensional statistical analysis, eg., factor analysis, principal component analysis, principal coordinate analysis, cluster analysis or correlation analysis providing complex information on the relationships between the individual morphophysiological and biochemical characteristics determined in the study.
Data availability statement: I think that the metabolomic data should be submitted to some public repository and the repository name and data accession number should be provided in the Data availability statement.
Formal comments on the text:
Results and Discussion, line 79: Add a comma between the words „79.56%“ and „respectively;“ line 80: Add a comma between the words „81.99%“ and „respectively“.
Results and Discussion, line 91: Modify the word forms „a decraese trend“ to „a decreasing trend…“
Figure 1 legend, line 94: Modify the word form „characters“ to „characteristics“.
Line 108: The reference „Serrander et al….“ – a reference number is missing. The reference has also be added to the Reference list where it is currently missing.
Line 183: Correct the typing error in the word „14 metabolites“.
Figure 5 legend, line 226: Ad dan explanation „C = control“ to the figure legend.
The legends to Figure 3, 4, and 6 miss important information on the software and algorithms used. Which software was employed for the cluster analysis presented in Figure 3 or for biological endpoint analysis provided in Figure 4??
Figure 6 legend is very brief. I think that it should be modified as follows: „Metabolic pathways as affected by sulfamethoxazole treatment. Blue indicates metabolites revealing differential accumulation between sulfamethoxazole-treated and untreated plants.“
In Figure 6 scheme, correct the typing error in the chemical name „phosphoenolpyruvic acid“.
Materials and methods, line 264: Add a comma both prior to and after the word „finally“ in the statement „…and, finally, injecetd into…“
Line 267: Add a comma between the words „250 °C“ and „respectively.“
Final recommendation: Reconsider after a major revision.

Author Response
Comments 1: The manuscript entitled „Comprehensive analysis of metabolites and biological endpoints provides new insights into the tolerance of wheat under sulfamethoxazole stress“ represents a useful study aimed at an investigation of the effects of sulfamethoxazole treatment on wheat response at physiological and biochemical (metabolite) levels. I think that the manuscript provides interesting novel data on sulfamethoxazole effects on wheat physiological and biochemical characteristics which are worth publishing in International Journal of Molecular Sciences.
Response 1: [Thank you very much for your recognition.]
Comments 2: However, I have several comments on the present manuscript which are provided below.
In Materials and methods, the wheat plants used for the experiments have to be properly characterized by providing the information on the wheat cultivar/genotype used and the source of seeds by providing the institution (genebank, breeding company) name, city, and country.
Response 2: [Thank you very much for your question. We have perfected the method.]
Comments 3: In Materials and methods, appropriate description of methodological approaches for the presenetd data has to be added, i.e., the description of methodology used for the construction of endpoint graphs provided in Figure 4 has to be provided including the software or any other online tools used.
Response 3: [Thank you very much for your question. We have perfected the method. Please refer to “3.4. Statistical Analysis”.]
Comments 4: In Materials and methods, section on Statistical analysis, the kind of a post-hoc test of multiple comparisons following ANOVA analysis has to be specified. The same information, i.e., the kind of statistical test used for the determination of significant differences, has to be added to the legends to Figure 1 and Figure 2.
Response 4: [Thank you very much for your question. We have perfected the method. Please refer to “3.4. Statistical Analysis”.]
Comments 5: In Materials and methods, section 3.3. Metabolites, the basic information on gas chromatographic column including the company (manufacturer) has to be added.
Response 5: [Thank you very much for your question. We have perfected the method. Please refer to “3.3. Metabolites”.]
Comments 6: In Results, I would recommend the authors to add some kind of multidimensional statistical analysis, eg., factor analysis, principal component analysis, principal coordinate analysis, cluster analysis or correlation analysis providing complex information on the relationships between the individual morphophysiological and biochemical characteristics determined in the study.
Response 6: [Thank you very much for your question. We have perfected these contents.]
Comments 7: Data availability statement: I think that the metabolomic data should be submitted to some public repository and the repository name and data accession number should be provided in the Data availability statement.
Response 7: [Thank you very much for your question. We've uploaded the metabome data to the Supplementary File.]
Comments 8: Formal comments on the text:
Results and Discussion, line 79: Add a comma between the words „79.56%“ and „respectively;“ line 80: Add a comma between the words „81.99%“ and „respectively“.
Results and Discussion, line 91: Modify the word forms „a decraese trend“ to „a decreasing trend…“
Figure 1 legend, line 94: Modify the word form „characters“ to „characteristics“.
Line 108: The reference „Serrander et al….“ – a reference number is missing. The reference has also be added to the Reference list where it is currently missing.
Line 183: Correct the typing error in the word „14 metabolites“.
Figure 5 legend, line 226: Ad dan explanation „C = control“ to the figure legend.
The legends to Figure 3, 4, and 6 miss important information on the software and algorithms used. Which software was employed for the cluster analysis presented in Figure 3 or for biological endpoint analysis provided in Figure 4??
Figure 6 legend is very brief. I think that it should be modified as follows: „Metabolic pathways as affected by sulfamethoxazole treatment. Blue indicates metabolites revealing differential accumulation between sulfamethoxazole-treated and untreated plants.“
In Figure 6 scheme, correct the typing error in the chemical name „phosphoenolpyruvic acid“.
Materials and methods, line 264: Add a comma both prior to and after the word „finally“ in the statement „…and, finally, injecetd into…“
Line 267: Add a comma between the words „250 °C“ and „respectively.“
Response 8: [Thank you very much for your question. We made changes to these details in the manuscript.]

Round 2
Reviewer 1 Report
Comments and Suggestions for Authors
Thank you for your thorough revision of the manuscript. I acknowledge that you have addressed the majority of the concerns raised in my previous review. The manuscript is now clearer and better structured, with improved interpretation of the metabolic and physiological results. I appreciate the effort made to enhance the quality of the figures and the overall presentation. However, I suggest two minor improvements : (1) please include a brief justification for the choice of sulfamethoxazole concentrations in the Materials and Methods section, supported by relevant references or environmental relevance; and (2) consider expanding the discussion with a short reflection on the potential ecological implications of sulfamethoxazole contamination in real agricultural systems. These additions would further strengthen the manuscript’s contribution to the field.
Bets regards
Author Response
Comments: Thank you for your thorough revision of the manuscript. I acknowledge that you have addressed the majority of the concerns raised in my previous review. The manuscript is now clearer and better structured, with improved interpretation of the metabolic and physiological results. I appreciate the effort made to enhance the quality of the figures and the overall presentation. However, I suggest two minor improvements : (1) please include a brief justification for the choice of sulfamethoxazole concentrations in the Materials and Methods section, supported by relevant references or environmental relevance; and (2) consider expanding the discussion with a short reflection on the potential ecological implications of sulfamethoxazole contamination in real agricultural systems. These additions would further strengthen the manuscript’s contribution to the field.
Response: [Thank you very much for your recognition. (1) The concentration of sulfamethoxazole is determined based on the environmental concentration of sulfamethoxazole and the concentration in related toxicological studies [21,39]. (2) SMX exhibits moderate persistence, with degradation rates influenced by factors such as pH, organic matter content, and microbial activity. Prolonged SMX exposure enriches ARGs in soil microbiomes, contributing to the spread of multidrug-resistant bacteria [39, 40]. The transfer of SMX and ARGs into the food chain via contaminated crops or water raises public health concerns. Additionally, the erosion of soil ecosystem services may compromise agricultural sustainability. SMX contamination in agriculture poses multifaceted ecological risks, from soil microbiome disruption to ARG dissemination. Future research should prioritize field-based studies to quantify long-term impacts and evaluate mitigation efficacy. Please refer to manuscript.]
Reviewer 2 Report
Comments and Suggestions for Authors
Dear Authors,
in the revsed manuscript, I still have a few comments on the presented results.
In Figure 1 and 2 legends, the kind of statistical test used for the determination of significant differences has to be specified in the figure legend.
In Figure 3, 4 and 5 legends, the kind of software and the algorithms used for the construction of heat maps (Figure 3) and other specialised graphs (Figure 4, 5) has to be given in the figure legends.
Final recommendation: Accept after a minor revision.
Author Response
Comments: in the revsed manuscript, I still have a few comments on the presented results.
In Figure 1 and 2 legends, the kind of statistical test used for the determination of significant differences has to be specified in the figure legend.
In Figure 3, 4 and 5 legends, the kind of software and the algorithms used for the construction of heat maps (Figure 3) and other specialised graphs (Figure 4, 5) has to be given in the figure legends.
Response: [Thank you very much for your recognition, again. We have placed the software and methods in the legend. Please refer to manuscript.]